# *Gordonia sputi* as an Arising Causative Agent of Bacteremia in Immunocompromised Comorbid Dialysis Patients—A Case Report

**DOI:** 10.3390/healthcare11142059

**Published:** 2023-07-19

**Authors:** Bozhidar Vergov, Andreana Angelova, Alexandra Baldzhieva, Yordan Kalchev, Georgi Tsochev, Marianna Murdjeva

**Affiliations:** 1Department of Medical Biology, Faculty of Medicine, Medical University of Plovdiv, 4000 Plovdiv, Bulgaria; 2Department of Dialysis Treatment, St George University Hospital, 6000 Plovdiv, Bulgaria; georgi.tsochev@mu-plovdiv.bg; 3Department of Medical Microbiology and Immunology “Prof. Dr. Elissay Yanev”, Faculty of Pharmacy, Medical University of Plovdiv, 4000 Plovdiv, Bulgaria; andreana.angelova@mu-plovdiv.bg (A.A.); alexandra.baldzhieva@mu-plovdiv.bg (A.B.); yordan.kalchev@mu-plovdiv.bg (Y.K.); mariana.murdzheva@mu-plovdiv.bg (M.M.); 4Laboratory of Microbiology, St George University Hospital, 6000 Plovdiv, Bulgaria; 5Research Institute at Medical University of Plovdiv, 4000 Plovdiv, Bulgaria; 6Laboratory of Clinical Immunology, St George University Hospital, 6000 Plovdiv, Bulgaria

**Keywords:** *Gordonia sputi*, dialysis, catheter-associated infection

## Abstract

Improvements in medical care have turned severe diseases into chronic conditions, but often their treatment and the use of medical devices are related to specific complications. Here, we present a clinical case of a long-term dialysis patient who was infected with a rare opportunistic infectious agent—*Gordonia sputi*. In recent years, the incidence of *Gordonia* spp. infections in immunocompromised patients with central venous catheters (CVC) has appeared to rise. The isolation and identification of *Gordonia* spp. are challenging and require modern techniques. In addition, the treatment is usually persistent and often results in CVC extraction, which is associated with further risk and costs for the patient. We also studied the alterations in the immune status of the patient caused by long-term renal replacement therapy and persistent hepatitis C virus infection. Antibiotic therapy and immunostimulation with Inosine pranobex lead to successful eradication of the infection without the need for CVC replacement.

## 1. Introduction

Long-term hemodialysis is a life-supporting procedure, but it has many complications for the patient. Chronic inflammation due to the contact of blood with artificial materials and uremia affects the immune status of the patient. Consequently, the leading causes of mortality in the hemodialysis population are cardiovascular diseases, withdrawal from renal replacement treatment, and infections [1,2,3,4].

Here, we present a clinical case of a hemodialysis patient with bacteremia caused by the opportunistic pathogen *Gordonia sputi*. Furthermore, we tried to investigate if long-term renal replacement therapy (more than 35 years) affected the immune status of the patient. 

*Gordonia* spp. are aerobic actinomycetes, found in soil and water, and were first described as a separate genus by Tsukamura in 1971 [5]. Some species are reported to cause infections in humans [6]. Even in rare incidences, their identification and treatment may be challenging. Colonization of medical devices appears to be a potential risk as some species are reported to adhere to and degrade rubber [7]. Often, successful treatment requires medical device extraction, which is related to higher risks and healthcare costs. 

## 2. Case Presentation

We report a 59-year-old female dialysis patient with a double-lumen tunneled venous catheter. Her dialysis treatment was initiated in 1985 due to four hypotrophic kidneys with superposed chronic glomerulonephritis. After multiple blood transfusions back then, the patient was infected with the hepatitis C virus (HCV). 

From November 2021, the patient reported malaise, weight loss, and febrile episodes of 37.5 °C to 38 °C the night after the dialysis procedure and the next day. No febrile episodes or chills during the procedure were noted. Multiple blood culture sets were tested, but no explicit bacterial agent was isolated. No leukocytosis was observed and only a slightly elevated C-reactive protein (CRP) was found. After every microbiology test, antibiotic treatment was applied, which resulted in temporary improvements in the symptoms. No clinical or laboratory findings, including echocardiography, were in favor of endocarditis. The patient refused a withdrawal of the dialysis catheter multiple times. 

In September 2022, upon a new relapse of symptoms, another blood culture set was sent to the microbiology laboratory. On the fifth day of incubation, it yielded gracile Gram-positive rods, which grew on sheep blood agar as small non-hemolytic white colonies. Phenotypic identification was performed using 4 h semi-automated biochemical testing with RapID™ CB PLUS (Thermo Scientific, Waltham, MA, USA) and the microorganism was identified as *Corynebacterium striatum.* Since no leukocytosis was found, and CRP was 37 °C, another blood culture sample was taken to rule out contamination or to confirm the result. Upon 4 days of incubation of the new blood culture set, the aerobic blood culture vial became positive. Direct microscopy revealed midsized actinomycete-like rods and upon cultivation at 37 °C, in an aerobic environment, small white colonies on sheep blood agar appeared. They were subjected to Matrix-Assisted Laser Desorption Ionization–Time-of-Flight Mass Spectrometry (MALDI-TOF MS, Vitek MS, bioMerieux, France) identification and the protein profiles obtained were characteristic of *Gordonia sputi* (Figure 1). The antimicrobial susceptibility testing (AST) according to EUCAST revealed the isolate was susceptible to vancomycin, gentamicin, linezolid, imipenem, ceftriaxone, and ciprofloxacin. After antibiotic treatment with gentamicin and meropenem for 21 days, all symptoms disappeared, and the improvement in the patient’s condition remained constant. On the second and the fourth week after the end of the antibiotic course, blood culture samples were negative. Meanwhile, on 22 July 2022, *Candida tropicalis* from a throat swab was isolated. The candida infection persisted for about 6 months despite the peroral fluconazole therapy. A timeline of the events is presented in Figure 2. 

In March 2023, the patient was referred to the clinical immunology unit for further investigation of the immune status. For the assessment of humoral immunity, the levels of serum IgA, IgG, IgM, and complement fractions C3 and C4 were measured by an automated immunoturbidimetry analyzer (BA200, Biosystems, Barcelona, Spain). An internal quality control study (using two levels of control serum offered by the manufacturer) and calibration were performed according to the manufacturer’s instructions (Biosystems, Barcelona, Spain). The values of serum immunoglobulins are expressed in µg/mL and C3 and C4 levels are expressed in g/L (Table 1). 

Laboratory investigations showed normal levels of the three immunoglobulins and the complement fractions. For the evaluation of the cellular immune status, lymphocyte subpopulation counts (LSc) were measured by 6-color TBNK reagent using Trucount Absolute Counting Tubes (BD, New Jersey, USA) in a peripheral venous blood sample within 2 h of blood draw by BD FACSCanto II, BD, USA, and the kit consisted of the following monoclonal antibodies: CD3-FITC (clone SK7), CD16-PE (clone B73.1), CD56-PE (clone NCAM16.2), CD45-PerCP-Cy5.5 (clone 2D1), CD4-PE-Cy7 (clone SK3), CD19-allophycocyanin (APC) (clone SJ25C1), and CD8-APC-Cy7 (clone SK1) (Figure 3).

Flowcytometric testing revealed an impairment of the cell-mediated immunity with lymphopenia and a decreased absolute number of immunocompetent CD3+ T cells, helper/inducer CD4+ T cells, and cytotoxic/suppressor CD8+ T cells, and borderline low B and NK (natural killer) cells. The CD4/CD8 T cell ratio was normal. (Table 2). 

Additionally, cytokine analysis was conducted using a human Th1/Th2/Th17 cytokine cytometric bead array kit (Cytometric Bead Array (CBA) Human Th1/Th2/Th17 Cytokine Kit, BD, USA), which allowed for the simultaneous detection of the IL-2, IL-4, IL-6, IL-10, TNF-a, IFN-ɣ, and IL-17A cytokines in serum. The altered function of immune cells resulted in an intriguing dysregulation of cytokine production characterized by elevated levels of Th2 cytokines (IL-4, IL-6, and IL-10) and slightly detectable Th1 (IFN-ɣ, TNF-α, and IL-2) and Th17 (IL-17A) cytokines (Table 3). 

## 3. Discussion

The clinical and laboratory changes consistent with infection were weakly manifested, probably because of the patient’s comorbidity and deprived immune status. Clinical presentation was additionally concealed by the empirical antibiotic courses and later by the superposed fungal infection. The refusal for catheter removal interfered with the prompt identification and eradication of the infection. 

Dialysis patients with central venous catheters are reported to have higher rates of mortality and complications, i.e., endocarditis, septic shock, and abscesses, compared to other vascular accesses: arterio-venous fistulas and vascular grafts. The same article pointed out that despite the many problems of catheters, their placement may be inevitable and, because of the profile of patients that begin hemodialysis, they are widely used, i.e., in older patients often with many comorbidities. The construction of an arterio-venous fistula and its maturation in these cases may be difficult [8]. 

The spectrum of the causative agents of hemodialysis-related infections is similar in cases of vascular access and catheter-related bacteremia. More than half of them are caused by Gram-positive bacteria, the most common of which is *S. aureus*, including methicillin-resistant *S. aureus (MRSA*). Coagulase-negative staphylococci (CoNS) are also common, predominantly *S. epidermidis* [9,10,11]. Approximately 25% of cases are caused by Gram-negative bacteria such as *Escherichia coli (E. coli), Pseudomonas aeruginosa, Enterobacter* spp., and *Klebsiella* spp. as well as *Proteus* spp. and fungi from the *Candida genus* [11,12].

With the increasing use of MALDI-TOF MS and 16S rRNA sequencing, bacteria previously not known to be associated with certain clinical syndromes have been newly identified. This particular patient population is also susceptible to opportunistic infections caused by rare pathogens such as *Gordonia* species. They are emerging pathogens in hemodialysis patients.

Little is known about the epidemiology of *Gordonia* spp. in general and its association with human diseases. The *Gordonia* genus has a complicated taxonomic history of several reclassifications. The picture is further complicated by the fact that identification is challenging and misidentification often occurs due to the close relation of other genera within the *Mycobacteriales* order, like *Dietzia* spp., *Corynebacterium, Nocardia* spp., *Rhodococcus* spp., and Tsukamurella spp., with *Gordonia* spp. Reports of human infections caused by *Gordonia* spp. are relatively rare when compared to other opportunistic pathogens of closely related taxonomic genera like *Nocardia* spp. and *Rhodococcus* spp. A bibliographic review indicates that the use of catheters for long-term intravenous access is a notable risk factor for bloodstream infections caused by *Gordonia* species. A recent report from France showed that *Gordonia* spp., including *G. sputi*, are indeed recovered from immunocompromised hosts like HIV-positive individuals, and individuals with malnutrition or long-term corticosteroid treatment, etc., but also when indwelling catheters of any type are present [13,14,15,16,17,18,19,20].

In our case, for the time between November 2021 and September 2022, the patient had multiple febrile episodes where conventional blood culture testing did not yield a definitive causative agent and the applied antimicrobial treatment had a temporary effect. The first possible causative agent in our patient, detected in September 2022, was identified by semi-automated biochemical testing (RapID CB Plus) as *Corynebacterium striatum*. The mentioned test does not include *Gordonia* spp. in its diagnostic spectrum. Because of this fact and due to the close relativity and overlapping of some morphological and biochemical characteristics of *Gordonia* spp. and its other neighboring genera, similar to the other authors, we cannot exclude the possibility for the first isolate to have been *Gordonia sputi* misidentified as *Corynebacterium striatum.* This goes to show that routine methods are insufficient and more complex and modern techniques are needed, e.g., proteomic analysis with mass-spectrometry or molecular genetic assays like polymerase chain reaction (PCR) or 16S ribosomal RNA sequencing. Lai et al. retrospectively re-evaluated 66 samples that were initially identified by conventional techniques as *Rhodococcus* spp. and found that when using the molecular method 15 of them were re-identified as *Gordonia* spp. [21].

Matrix-assisted laser desorption/ionization-time-of-flight (MALDI-TOF) mass spectrometry (MS)—MALDI-TOF MS—is a method for rapid and accurate identification that is becoming increasingly available in the clinical microbiology laboratory. It is a technique that is based on producing ionized particles from a bacterial–matrix mixture, which are then separated according to their mass-to-charge ratio. A unique spectrum is generated, which is in turn compared to a database of known and validated microorganisms. MALDI-TOF MS is capable of identifying a wide spectrum of microorganisms including Gram-positive and Gram-negative bacteria, mycobacteria, yeast, and molds. Often, the accuracy is comparable to molecular methods of identification such as 16S rRNA gene sequencing [22,23,24].

Thanks to the MALDI-TOF MS method, we managed to elucidate the etiology and initiate appropriate treatment. We consider MALDI-TOF MS a cheap, rapid, and reliable method for the accurate identification of *G. sputi* on a species level. Precise identification is also important in providing additional information about the association of *G. sputi* and extending our knowledge on the epidemiology of *G. sputi* and its role as an opportunistic pathogen. 

Also, it is crucial to underline the significance of opportunistic isolates such as *Gordonia sputi* in immunosuppressed patients. For an adequate immune response towards infectious agents, a sufficient number of immune-competent cells are needed and, in our patient, the flowcytometry testing indisputably confirmed lymphopenia with suppression of major subsets of cells with the most remarkable decrease in the CD3+ T-cells count. This is an important factor supporting the invasiveness of infections. Such findings regarding the cellular immune status are also present in various studies [25,26].

Another reason for the impaired immune response is the dysregulation of cytokine production resulting in an imbalanced differentiation of Th lymphocytes to Th1 or Th2 cells. Each of the corresponding subpopulations secretes distinct cytokines—Th1 cytokines are IL-2, TNF-α, IFN-γ, etc., while IL-4, IL-6, IL-10, etc. belong to the group of Th2 cytokines [27]. Our patient’s immune status demonstrates an impairment of cell-mediated immunity, which is sustained by Th1 cells (slightly detectable levels of Th1 cytokines) with preserved humoral immunity marked by normal levels of total immunoglobulins and complement fractions C3 and C4 sustained by Th2 cells (increased levels of Th2 cytokines). It is known that IL-4 as well as IL-10 enhance Th2 and inhibit Th1 development [28,29]. According to some other authors, the levels of Th2 cytokines in hemodialyzed patients are increased [27] [30], which corresponds to the results in our patient. A study by Szabo et al. demonstrates that IL-4 inhibits the expression of the signal-transducing β2 subunit of the IL-12 receptor and thus the ability of the latter to induce a Th1 response [31]. Additionally, both IL-4 and IL-10 possess direct anti-inflammatory properties [32,33,34,35]. The hindrance of Th1 cytokines may result in complex defects of cell-mediated effector functions, including the phagocytic elimination of infectious agents, macrophage inflammatory cytokine production, and natural killer cell– and CD8+ T-cell–mediated cytotoxicity [36]. Moreover, chronic hepatitis C infection is associated with impaired function of helper/inducer CD4+ T cells and cytotoxic/suppressor CD8+ T cells and an overactive Th2 immune response [37,38,39,40,41,42] Thus, the comorbidities presented by chronic liver infection, long-term hemodialysis, and cancer in her adolescence correspond to the immune suppression in this patient. 

Immune stimulation is an important therapeutic measure in such patients. Due to the intact levels of total immunoglobulins in this case, the administration of intravenous immunoglobulin (IVIG) was not taken into consideration. A therapeutic approach for this patient is Inosine pranobex (IP), commonly known as Isoprinosine, which is known to enhance T-cell lymphocyte proliferation and the activity of NK cells, leading to the restoration of the deficient responses in immunosuppressed patients with advantageous effects also on HCV infection [43].

## 4. Conclusions

The clinical presentation of catheter-associated bacteriemia in polymorbid hemodialysis patients may be vague because of a depressed immune system. Prophylaxis of infections by these patients is crucial because of the many life-threatening complications. These patients must be closely followed-up and even when mild symptoms are presented physicians should be encouraged to take blood cultures. 

Given the current rise in immunocompromised hosts as well as the prominent increase in venous catheter use, it is crucial to precisely identify *Gordonia* spp. on a species level. This, in turn, will not only help us better understand the epidemiology of *G. sputi* infections but also aid in improving strategies and optimizing the treatment guidelines. Increased awareness among clinicians, including clinical microbiologists, would be beneficial to high-risk populations and public health in general. This case illustrates that some rapid commercially available microbiological identification systems may provide inaccurate results, and the precise identification to the species level can be achieved by assays that are more complex but still accessible for most laboratories like MALDI-TOF mass spectrometry. We consider MALDI-TOF mass spectrometry a reliable alternative to molecular methods that can provide rapid, cheap, and accurate identification of *Gordonia* spp. on a species level. 

We may also conclude that the treatment of immunocompromised comorbid hemodialysis patients should always include a consideration of the constant risk of opportunistic infections. Their management should involve protective measures against the latter, prophylaxis of fungal infections, and appropriate immune stimulation. 

## Figures and Tables

**Figure 1 healthcare-11-02059-f001:**
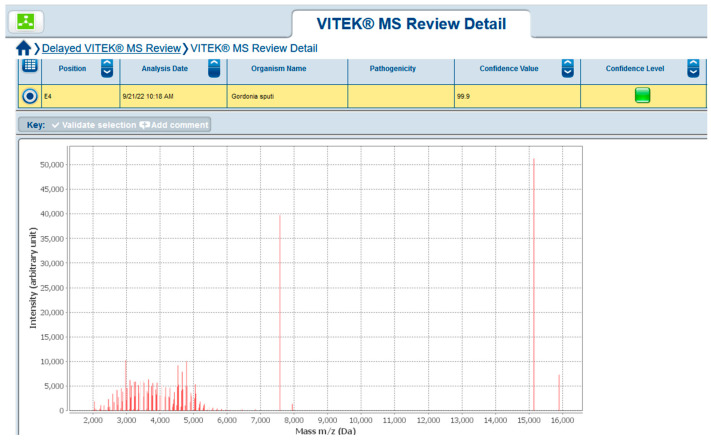
*Gordonia sputi* MALDI-TOF MS result.

**Figure 2 healthcare-11-02059-f002:**
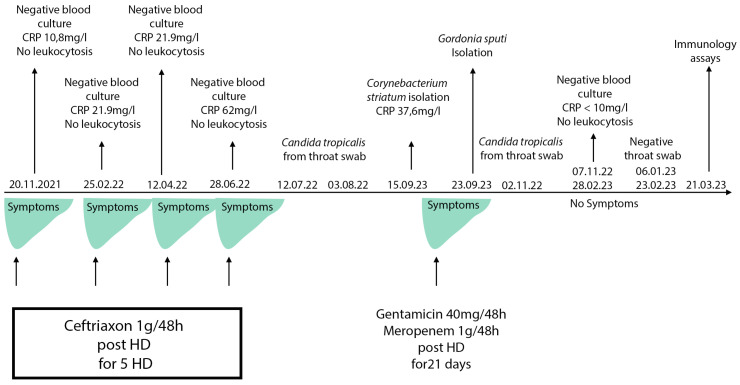
Timeline of the clinical case events. HD: hemodialysis. CRP (C-reactive protein) reference range: 0–10 mg/L.

**Figure 3 healthcare-11-02059-f003:**
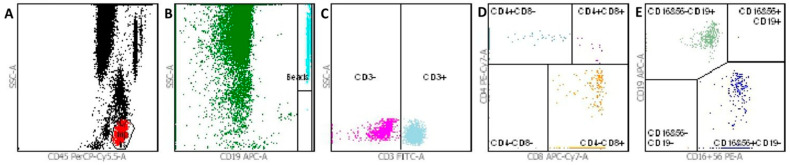
Flow cytometry gating strategy: FACS dot plot on BD FACS Canto II gating CD4 and CD8 T cells. Panel (**A**) depicts CD45 + lymphocytes detected in the dot plot of CD45 PerCP-Cy5.5-A vs. SSC-A. Panel (**B**) shows the CD19 APC-A vs. SSC-A dot plot with BD Trucount absolute count bead events. Panel (**C**) depicts CD3 + T cells in the dot plot of CD3 FITC-A vs. SSC-A. In the CD8 APC-Cy7-A vs. CD4 PE-Cy7-A dot plot, panel (**D**) depicts suppressor/cytotoxic (CD4-CD8 +) and helper/inducer (CD4 + CD8 -) T lymphocytes. Panel (**E**) illustrates the natural killer subset (NK cells) identified as CD3– and CD16+ and/or CD56+.

**Table 1 healthcare-11-02059-t001:** Results from the evaluation of the serum immunoglobulins and complement fractions.

Indicator	Measurement Unit	Result	Reference Range
IgG	µg/mL	14,073	5000–17,000
IgM	µg/mL	940	400–2500
IgA	µg/mL	2458	200–3000
C3	g/L	1.23	0.90–1.80
C4	g/L	0.26	0.10–0.40

**Table 2 healthcare-11-02059-t002:** Results from immunophenotyping of lymphocyte subpopulations performed on FACSCanto II Clinical Flow Cytometry System, situated in Medical Microbiology and Immunology Department of Medical University-Plovdiv according to a standardized procedure and using commercial TBNK-multitest reagent kit and national age-adjusted reference ranges of lymphocyte subsets.

Indicator	Unit of Measurement	Results	Reference Range
Absolute number of leukocyte subpopulation (number of cells × 10^9^/L)			
Lymphocytes	10^9^/L	0.91	1.0–2.8
Total CD3+ T cells	10^9^/L	0.64	1.0–2.0
T help-induc CD3+CD4+	10^9^/L	0.45	0.6–1.4
T suppr-cytotoxic CD3+CD8+	10^9^/L	0.17	0.3–1.0
Total B cells CD19+	10^9^/L	0.10	0.1–0.4
NK cells CD3-CD56+	10^9^/L	0.17	0.1–0.6
Percentage of leukocyte subpopulation			
Total CD3+ T cells	%	70.34	61–85
T help-induc CD3+CD4+	%	48.75	34–59
T suppr-cytotoxic CD3+CD8+	%	18.80	19–36
Total B cells CD19+	%	11.29	6–15
NK cells CD3-CD56+	%	18.29	7–26
Index CD4/CD8	2.59	2.59	0.9–3.0

**Table 3 healthcare-11-02059-t003:** Results from measurement of cytokine profile Th1/Th2/Th17 by BD CBA Th1/Th2/Th17 kit.

Cytokine	Concentration (pg/mL)
IFN gamma	0
TNF alpha	5
IL-2	0
IL-4	6
IL-6	61
IL-10	9
IL-17A	15

## Data Availability

The raw clinical and laboratory data associated with the current study are available from the corresponding author, without undue reservation on a reasonable request.

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
