# Peer review of "Gordonia sputi as an Arising Causative Agent of Bacteremia in Immunocompromised Comorbid Dialysis Patients—A Case Report"

_healthcare, 2023, doi:10.3390/healthcare11142059_

Round 1

Reviewer 1 Report

The subject of the submitted manuscript is of interest to the research field and presents a very interesting case report of a haemodialysis patient with a double lumen tunnelled venous catheter who appears to have been infected by the gordonia spp. bacteria.

Overall, this is a very well written and structured document, that reads clearly and thoroughly presents this clinical case. An interesting and detailed discussion is presented, which has been written following good scientific standards. For this reason, the authors will only find some very minor comments in the attached document. I have no further comments on the submitted manuscript, using the opportunity to congratulate and encourage the authors to carry on with the interesting work that is being produced.

Author Response

Dear Reviewr,

Here are our corrections according to your points

Point 1: HCV has not been defined in abstract. Perhaps consider writing in full in the abstract and defining it in section 2. The 1st time it is mentioned.

Response 1: We accepted your suggestion and edited the manuscript according to it. In the abstract on line 23 it is Hepatitis C virus infection and on line 46 we added : After multiple blood transfusions back then, the patient was infected with the hepatitis C virus (HCV)

Point 2: Lines 32-33: The main reference used is slightly dated (1991) .Please consider inserting more recent evidence.

 Response 2:  We inserted the following two  references  from 2020 and 2022:

Annual Data Report | USRDS. (n.d.). Retrieved July 8, 2023, from https://usrds-adr.niddk.nih.gov/2022/end-stage-renal-disease/6-mortality

Siga, M. M., Ducher, M., Florens, N., Roth, H., Mahloul, N., Fouque, D., & Fauvel, J. P. (2020). Prediction of all-cause mortality in haemodialysis patients using a Bayesian network. Nephrology Dialysis Transplantation, 35(8), 1420–1425. https://doi.org/10.1093/NDT/GFZ295

 It appears that together with cardiovascular diseases and infections, the withdraw from hemodialysis is among the main mortality causes. The information is added to the manuscript on line 32.     

Point 3: Line 46: Replace tunneled by tunnelled .  

Response 3: On line 46 tunneled is replaced by tunnelled
Best regards,

Bozhidar Vergov

Reviewer 2 Report

I recommend the authors to improve the conclusions since they do not highlight the findings of the case and the importance of recognizing Gordonia sputi as an opportunistic agent in immunocompromised patients.

The manuscript is well written, however I recommend the authors to improve the description of the methods used.

Author Response

Dear Reviewer,

Please find attached our response to your points.

Best regards,
Bozhidar Vergov

Reviewer 3 Report

The authors are very superficial in the introduction and discussion.

What are the leading infectious agents in hemodialysis, and why is Gordonia sputi relevant?

The authors must show figures demonstrating their findings and making their work attractive.

They must show a workflow on the main findings and dates of diagnosis (symptoms and laboratory data) of the patient, including treatment and outcome.

The authors mention various inflammatory markers (cellular and humoral). However, they are very qualitative in their results; the writing would be enriched if they showed the quantitative data of each marker.

The authors must describe and give due importance to the MALDI-TOF method and how they could identify the causative agent of the infection thanks to it.

Citation 11 is incomplete; you should look for more recent sources, perhaps there is not much about Gordonia, but there is about the use of mass spectrometry and mainly MALDI-TOF as a method of diagnosing infections.

The work is publishable. However, a restructuring of this.

Line 145 check the punctuation.

Author Response

Dear Reviewer,
Please, find attached our responses to your comments.
Best regards,
Bozhidar Vergov,

Round 2

Reviewer 3 Report

The article is ready

None